# Predictors of perinatal death in the presence of missing data: A birth registry-based study in northern Tanzania

**Innocent B. Mboya**[1,2]*, **Michael J. Mahande**[2], **Joseph Obure**[3], **Henry G. Mwambi**[1]

**1** School of Mathematics, Statistics & Computer Science, University of KwaZulu-Natal, Pietermaritzburg, South Africa, **2** Department of Epidemiology and Biostatistics, Institute of Public Health, Kilimanjaro Christian Medical University College (KCMUCo), Moshi-Tanzania, Tanzania, **3** Department of Obstetrics and Gynecology, Kilimanjaro Christian Medical Center (KCMC), Moshi-Tanzania, Tanzania

* ib.mboya@gmail.com

**Data Availability Statement:** The KCMC medical birth registry data contains potentially identifying and sensitive patient information. This has also been stipulated by the local Institutional Review

## Abstract

### Background

More than five million perinatal deaths occur each year globally. Despite efforts put forward during the millennium development goals era, perinatal deaths continue to increase relative to under-five deaths, especially in low- and middle-income countries. This study aimed to determine predictors of perinatal death in the presence of missing data using birth registry data from Kilimanjaro Christian Medical Center (KCMC), between 2000–2015.

### Methods

This was a retrospective cohort study from the medical birth registry at KCMC referral hospital located in Moshi Municipality, Kilimanjaro region, northern Tanzania. Data were analyzed using Stata version 15.1. Multiple imputation by fully conditional specification (FCS) was used to impute missing values. Generalized estimating equations (GEE) were used to determine the marginal effects of covariates on perinatal death using a log link mean model with robust standard errors. An exchangeable correlation structure was used to account for the dependence of observations within mothers.

### Results

Among 50,487 deliveries recorded in the KCMC medical birth registry between 2000–2015, 4.2% (95%CI 4.0%, 4.3%) ended in perinatal death (equivalent to a perinatal mortality rate (PMR) of 41.6 (95%CI 39.9, 43.3) deaths per 1,000 births). After the imputation of missing values, the proportion of perinatal death remained relatively the same. The risk of perinatal death was significantly higher among deliveries from mothers who resided in rural compared to urban areas (RR = 1.241, 95%CI 1.137, 1.355), with primary education level (RR = 1.201, 95%CI 1.083, 1.332) compared to higher education level, with <4 compared to ≥4 antenatal care (ANC) visits (RR = 1.250, 95%CI 1.146, 1.365), with postpartum hemorrhage (PPH) (RR = 2.638, 95%CI 1.997, 3.486), abruption placenta (RR = 4.218, 95%CI 3.438, 5.175), delivered a low birth weight baby (LBW) (RR = 4.210, 95%CI 3.788, 4.679), male child (RR

Board of KCMC hospital and the National Ethics Committee in Norway when establishing this birth registry. Permission to use the data in this study was made through the Kilimanjaro Christian Medical University College Research and Ethics Review Committee, and received an approval number 2424. The authors do not have the legal right to share the data publicly. All data requests can be sent directly to the Executive Director of the KCMC referral hospital, P. O. Box 3010, Moshi, Tanzania, Email: drgmasenga@yahoo.com or through the corresponding author.

**Funding:** This work was supported through the DELTAS Africa Initiative Grant No. 107754/Z/15/ZDELTAS Africa SSACAB. The DELTAS Africa Initiative is an independent funding scheme of the African Academy of Sciences (AAS)'s Alliance for Accelerating Excellence in Science in Africa (AESA) and supported by the New Partnership for Africa's Development Planning and Coordinating Agency (NEPAD Agency) with funding from the Wellcome Trust (Grant No. 107754/Z/15/Z) and the UK government. The views expressed in this publication are those of the author(s) and not necessarily those of AAS, NEPAD Agency, Wellcome Trust or the UK government. The Norwegian Council for Higher Education's Program for Development Research (NUFU) funded the establishment of the birth registry at the Kilimanjaro Christian Medical Center (KCMC). The funders had no role in study design, data collection, and analysis, decision to publish, or preparation of the manuscript.

**Competing interests:** The authors have declared that no competing interests exist.

= 1.090, 95%CI 1.007, 1.181), and were referred for delivery (RR = 2.108, 95%CI 1.919, 2.317). On the other hand, deliveries from mothers who experienced premature rupture of the membranes (PROM) (RR = 0.411, 95%CI 0.283, 0.598) and delivered through cesarean section (CS) (RR = 0.662, 95%CI 0.604, 0.724) had a lower risk of perinatal death.

## Conclusions

Perinatal mortality in this cohort is higher than the national estimate. Higher risk of perinatal death was associated with low maternal education level, rural residence, <4 ANC visits, PPH, abruption placenta, LBW delivery, child's sex, and being referred for delivery. Ignoring missing values in the analysis of adverse pregnancy outcomes produces biased covariate coefficients and standard errors. Close clinical follow-up of women at high risk of experiencing perinatal death, particularly during ANC visits and delivery, is of high importance to increase perinatal survival.

## Introduction

Perinatal death refers to the number of stillbirths (pregnancy loss that occurs after seven months of gestation and before birth) and early neonatal deaths (deaths of live births within the first seven days of life)[1, 2]. Perinatal and maternal health are closely linked; hence perinatal mortality is used as an essential indicator to monitor maternal health status and quality of antenatal, intrapartum, and newborn care[1, 3–5]. Globally, more than five million perinatal deaths occur each year[5]. Children face the highest risk of dying in their first month of life at a global rate of 18 deaths per 1,000 births[6]. Globally, 2.5 million children died in the first month of life in 2018–7,000 deaths every day[6]. The patterns of these deaths are similar to the patterns for maternal deaths, the majority occurring in developing countries[1]. In Tanzania, between the years 2004–2005 and 2015–16, the under-five mortality rate was reported to have declined from 112 to 67 deaths per 1,000 births. The country has, however, witnessed an increase in the number of stillbirths (from 143 to 187), the number of early neonatal deaths (from 156 to 214) as well as perinatal mortality rate (from 36 to 39) deaths per 1,000 births, respectively[2].

The risk factors for stillbirths and early neonatal deaths are closely linked, and examining just one or the other is reported to bias the true level of mortality around delivery [2, 7]. The risk of perinatal mortality has been associated with preterm birth, shorter birth interval (<24 months), congenital anomalies, previous history of early neonatal death, low birth weight, maternal anemia, placental abruption, ruptured uterus, systemic infections/sepsis, pre-eclampsia, eclampsia, obstetric hemorrhage, having a home delivery, fetal growth restrictions and maternal infections such as syphilis and malaria[3, 4, 8–13]. As with other adverse maternal outcomes, perinatal deaths recur in subsequent pregnancies[7, 14, 15]. Although these factors may be common across low, middle- and high-income countries, they are likely to differ depending on the context or country-specific conditions such as availability of quality obstetric and newborn care services at different levels of care.

Despite challenges in the coverage and content of antenatal care[16], the WHO recommends a minimum of eight contacts for antenatal care (ANC) that can reduce perinatal deaths by up to eight per 1000 births when compared to a minimum of four visits[17]. Early identification and management of women with complications have also been recommended to

improve maternal and perinatal outcomes[12]. Informed interventions are therefore crucial to accelerate progress towards achieving the second indicator of the third sustainable development goal, i.e., by 2030, end preventable deaths of newborns and children under five years of age, with all countries aiming to reduce neonatal mortality to at least as low as 12 per 1,000 live births and under-5 mortality to at least as low as 25 per 1,000 live births[18]. These interventions should consider context-specific factors that can better explain the risk of perinatal deaths. As perinatal mortality rate is increasing in Tanzania, this study was aimed to determine current trends and associated factors from a maternally linked medical birth registry at KCMC referral hospital in northern Tanzania, comparing results before and after imputation of missing values.

Missing data is a common problem that occurs in almost all medical and epidemiological research[19–23]. Hospital-based longitudinal studies are also facing the same problem. Individuals with missing data may differ from those with no missing data in terms of the outcome of interest and prognosis in general[22]. Previous studies assessing predictors of perinatal death have adopted simple methods such as complete case analysis or available case analysis hence ignoring important information about missing data. Ignoring missing data in statistical analysis often produces biased and inefficient estimates of association[20, 22], especially when data are missing at random [24]. This study aimed to determine predictors of perinatal death accounting for missing values.

## Materials and methods

### Study design and participants

This study utilized data from the KCMC Medical Birth Registry, which contains maternally linked cohort data since the year 2000. KCMC is a national zonal referral hospital located in Moshi Municipality, Kilimanjaro Region, Northern Tanzania. The region has a total of seven districts, including the Moshi Municipality and Moshi rural district, where most of the births come from. This means that the main catchment area of the hospital is the local population. The hospital also admits referred cases from the rest of the regional districts as well as six other regions; Arusha, Kilimanjaro, Manyara, Tanga, Dodoma, and Singida [25]. The total fertility rate in the Kilimanjaro region was estimated to be 3.4 children [2].

The KCMC medical birth registry is located within hospital grounds at the Reproductive and Child Health Centre. The registry was established to secure a working system for medical birth registration and provide research data on the reproductive health of women and provide data for monitoring of perinatal health and quality of care at KCMC, among other purposes [25, 26]. The study population was women who delivered singleton babies. The study considered all deliveries recorded from January 2000 to December 2015, a total of 55,003 deliveries from 43,084 mothers aged 15–49 years. We excluded 3,316 multiples gestations to avoid over-representation of high-risk pregnancies [27]. We further excluded 49 records missing hospital numbers (i.e., unique identification number used to link mothers and their subsequent births) and 791 observations with a mismatch between dates of births of children from the same mother or were of unknown sequence (i.e., whether was a singleton or multiple births). We, therefore, analyzed data for 50,847 recorded deliveries born from 41,498 mothers (Fig 1).

### Data collection methods

A detailed description of the data collection procedure and data collected for the birth registry have been previously published[14, 28, 29]. Briefly, birth data at KCMC have been recorded using a standardized questionnaire. Specially trained project midwives did data collection, and mothers interviewed within the first 24 hours after birth given a normal delivery or on the

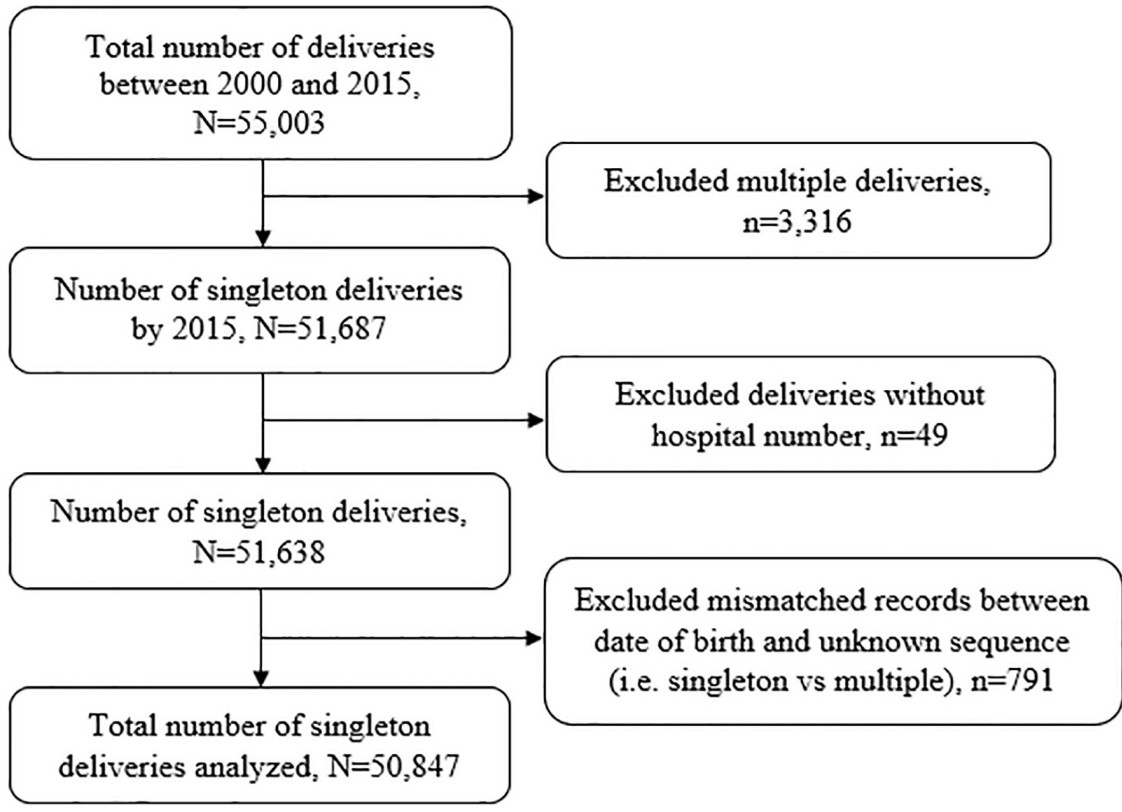

**Fig 1. Schematic diagram showing the number of participants, KCMC medical birth registry, 2000–2015.**

second or third day in case of complicated deliveries depending on their condition, following informed consent. Although the printed questionnaires are in the English language, the Project Midwives performing the interviews are well versed in English, Swahili, and one other tribal language. Patient files and antenatal care cards were used for verification of interview data and to extract additional information. Information of neonates admitted in the neonatal intensive care unit were recorded separately in the neonatal registry form and later linked with mother's information during entry in the birth registry using unique identification numbers. Furthermore, a unique identification number was assigned to each woman at first admission and used to trace her medical records at later admissions.

### Study variables and variable definitions

The response variable was perinatal death, which comprises stillbirths (pregnancy loss that occurs after seven months of gestation) and early neonatal death (death of live births within the first seven days of life)[1, 2]. We coded perinatal death as binary, i.e., 'yes' if death occurred during the perinatal period and 'no' if otherwise. The perinatal mortality rate is calculated as the number of perinatal deaths per 1,000 pregnancies of seven or more months' duration[2]. The reader should note that this outcome captured deaths that happened in the health facility only. There were no mechanisms in place to follow-up deaths occurring outside the health facility.

Independent variables included maternal background characteristics such as age categories (15–19, 20–34, 35–39 and 40+) in years, area of residence (rural vs. urban), highest education level (none, primary, secondary and higher), marital status (single, married and widow/

divorced), occupation (unemployed, employed and others), whether referred for delivery or not and the number of antenatal care visits (<4 and ≥4 visits). Maternal health before and during pregnancy; maternal body mass index (BMI) categorized as underweight (<18.5 Kg/m$^2$), normal weight (18.5–24.9 Kg/m$^2$), overweight (25–29.9 Kg/m$^2$) and obese (≥30 Kg/m$^2$). Alcohol consumption during pregnancy, maternal anemia, malaria, systemic infections/sepsis, and pre-eclampsia/eclampsia categorized as yes or no, with yes indicating the occurrence of these outcomes. Maternal HIV status was categorized as positive or negative. Information concerning delivery included complications during delivery, i.e., premature rupture of the membranes (PROM), postpartum hemorrhage, placenta previa, and placenta abruption categorized as yes and no. Gestational age at birth was estimated based on the date of the last menstrual period and was recorded in whole weeks. Preterm birth included any birth before 37 completed weeks of gestation[1, 30]. Newborn characteristics included sex (whether male or female), and low birth weight, defined as an infant birth weight of less than 2500g[31].

## Statistical analysis

We analyzed data using STATA version 15.1 (StataCorp LLC, College Station, Texas, USA). Numeric variables were summarized using means and standard deviations while categorical variables using frequency and percentages. We used the Chi-square test to compare the proportion of perinatal death across different levels of explanatory variables. Data analysis involved three stages; the first stage being complete case analysis, followed by imputation of missing values in both the outcome and the covariates, and finally, analysis of the imputed dataset. Results are compared before and after imputation to assess the precision of estimates.

Two separate models were fit before and after imputation of missing values. The first was generalized estimating equations (GEE) model with binomial family, logit link, exchangeable correlation structure, and robust variance estimator (Model I). The second was GEE log-linear regression model with the Poisson family, log link, an exchangeable correlation structure, and a robust variance estimator (Model II). The GEE models, often called population average or marginal models describes changes in the population mean given changes in covariates, while accounting for within-cluster correlation of observations [32]. These models are an extension to generalized linear models to longitudinal data, directly modelling the mean response, at each occasion using an appropriate link function [33]. The estimation procedures are not likelihood-based but uses a series of estimating equations. Odds Ratio (OR) and Relative risk (RR) with their corresponding 95% confidence intervals (CIs) were used to determine the strength of association at 5% significant level, respectively. Incomplete data were assumed to be missing at random (MAR) where the probability of data being missing does not depend on the unobserved data, conditional on the observed data [22, 34–36]; hence the variables in the dataset were used to predict missingness. Stepwise regression was used for variable selection, and it included a consecutive assessment of the effect of adding or removing different variables from the models.

Furthermore, for the analysis of missing data, we assumed a nonmonotone pattern of missingness in which some subject values were observed again after a missing value occurs [35, 36]. Multiple imputation is a commonly used method to deal with missing data, which accounts for the uncertainty associated with missing data[22, 24, 36]. Under a nonmonotone pattern of missingness, it is recommended to use the chained equations (also referred to as fully conditional specification (FCS) [21, 23]) or the Markov Chain Monte Carlo (MCMC) method to impute missing values [36]. We, therefore, used multiple imputation by FCS to handle missing data in this study. This technique is a powerful and statistically valid method for creating imputations in large datasets, which include both categorical and continuous variables [20, 21,

23, 34]. After declaring the preferred data structure using *mi set mlong* command, the *mi impute chained* function, part of *mi* package in Stata, implemented this technique, referred to in this software as Multiple Imputation by Chained Equations (MICE) approach [37–39]. Before imputation, *mi register imputed* command registered variables for the imputation model. Interested readers can find more about the *mi* command by typing *help mi* in their Stata command window.

Maternal age and education level were imputed using *ologit* command for ordinal response variables, while maternal occupation, marital status, and BMI (because normal weight (18.5– 24.9 Kg/m$^2$) was a reference category) using *mlogit* for multinomial distribution. The rest of the variables were binary, and so imputed using the *logit* command. Pre-eclampsia/ eclampsia, anemia during pregnancy, malaria, sepsis/ systemic infections, PROM, PPH, abruption placenta, and placenta previa did not contain any missing values, hence used as auxiliary variables in the imputation model. The imputation model generated 20 imputed datasets after 500 iterations (imputation cycles). A random seed of 5000 was specified for replication of imputation results each time this analysis was performed[36, 37]. We repeated similar procedures when imputing data for stratified analysis (i.e., by referral status), and determining the independent predictors of stillbirth after excluding early neonatal deaths. For the imputation model stratified by referral status, we imputed the data conditional on referral status (i.e., using *"if"* other than the *"by"* option in the Stata *mi impute chained* command options). After the imputation of missing values, we performed the analysis preceding with the "*mi estimate:*" command.

### Ethical consideration

This study was approved by the Kilimanjaro Christian Medical College Research Ethics and Review Committee (KCMU-CRERC) with approval number 2424. For practical reasons, since the interview was administered just after the woman had given birth, consent was given orally. The midwife-nurse gave every woman oral information about the birth registry, the data needed to be collected from them, and the use of the data for research purposes. Women were also informed about the intention to gather new knowledge, which will, in turn, benefit mothers and children in the future. Participation was voluntary and had no implications on the care women would receive. Following consent, mothers were free to refuse to reply to single questions. For privacy and confidentiality, unique identification numbers were used to both identity and then link mothers with child records. There was no any person-identifiable information in any electronic database, and instead, unique identification numbers were used. Necessary measures were taken by midwives to ensure privacy during the interview process.

### Results

#### Maternal background characteristics at first birth

The mean (SD) age of 41,498 mothers was 27 (6) years. More than three quarters (76.9%) were aged between 20–34 years, 55.9% resided in urban areas, 55.7% had primary education level, and 85.8% were married. The prevalence of underweight, overweight, and obese was 5.9%, 26.4%, and 11.1%, respectively. Twenty-eight percent of mothers drank alcohol during pregnancy (Table 1).

#### Obstetric care characteristics and complication

Malaria was the most common (13.5%) disease in this cohort. The proportion of pre-eclampsia/eclampsia was 4.1%, while that of HIV was 5.3%. About 30% of all deliveries had <4 ANC visits, 11.1% were delivered preterm (<37 weeks of gestation), 10.9% with low birth weight,

Table 1. Maternal background characteristics at first birth (N = 41,498).

| Characteristics | Frequency | Percent (95%CI) |
|---|---|---|
| **Age groups (years)*** | | |
| 15–19 | 4,250 | 10.3 (10.0, 10.6) |
| 20–34 | 31,866 | 76.9 (76.5, 77.3) |
| 35–39 | 4,201 | 10.1 (9.9, 10.4) |
| 40+ | 1,101 | 2.7 (2.5, 2.8) |
| **Area of residence*** | | |
| Rural | 18,244 | 44.1 (43.6, 44.6) |
| Urban | 23,137 | 55.9 (55.4, 56.4) |
| **Highest education level*** | | |
| None | 856 | 2.1 (1.9, 2.2) |
| Primary | 23,081 | 55.7 (55.3, 56.2) |
| Secondary | 4,895 | 11.8 (11.5, 12.1) |
| Higher | 12,582 | 30.4 (29.9, 30.8) |
| **Occupation*** | | |
| Unemployed | 9,386 | 22.8 (22.4, 23.2) |
| Employed | 28,973 | 70.3 (23.8, 24.6) |
| Others | 2,865 | 6.9 (6.7, 7.2) |
| **Marital Status*** | | |
| Single | 5,774 | 14.0 (13.6, 14.3) |
| Married | 35,468 | 85.8 (85.5, 86.1) |
| Widowed/Divorced | 99 | 0.2 (0.2, 0.3) |
| **Body mass index categories*** | | |
| Underweight (<18.5) | 1,671 | 5.9 (5.6, 6.1) |
| Normal weight (18.5–24.9) | 16,164 | 56.6 (56.0, 57.2) |
| Overweight (25–29.9) | 7,542 | 26.4 (25.9, 26.9) |
| Obese (> = 30) | 3,184 | 11.1 (10.8, 11.5) |
| **Drink alcohol during this pregnancy *** | | |
| Yes | 11,490 | 28.0 (27.6, 28.5) |
| No | 29,513 | 71.9 (71.5, 72.4) |

*Frequencies do not tally to the total due to missing values in these variables

and 34% delivered through Caesarean Section (CS). Almost a quarter (23.5) were referred for delivery. More than half (51.7%) of these deliveries were males (Table 2).

## Distribution of missing values in the KCMC medical birth registry

Table 3 summarizes frequencies and percentages of missing values for the variables with missing information in this study. Maternal BMI and HIV status had the highest proportions of missing values, 31.3%, and 23.7%, respectively. These two variables contributed to over half (55%) of all missing values in the dataset. The prevalence of missing values was about 8% in the gestational age at birth variable and 3.7% on referral status. The perinatal status (primary outcome) contributed only 0.2% of missing values in this dataset.

## Perinatal status by maternal characteristics during pregnancy and delivery

Among 50,724 deliveries with complete records on perinatal status in the KCMC medical birth registry between 2000–2015, 4.2% (95%CI 4.0%, 4.3%) ended in perinatal death

**Table 2. Diseases and complications during pregnancy and delivery (N = 50,847).**

| Characteristics | Frequency | Percent (95%CI) |
|---|---|---|
| **Pre-eclampsia/eclampsia** | | |
| No | 48,779 | 95.9 (95.8, 96.1) |
| Yes | 2,068 | 4.1 (3.9, 4.2) |
| **Anaemia** | | |
| No | 50,054 | 98.4 (98.3, 98.5) |
| Yes | 793 | 1.6 (1.5, 1.7) |
| **Malaria** | | |
| No | 43,961 | 86.5 (86.2, 86.8) |
| Yes | 6,886 | 13.5 (13.2, 13.8) |
| **Infections** | | |
| No | 49,982 | 98.3 (98.2, 98.4) |
| Yes | 865 | 1.7 (1.6, 1.8) |
| **HIV Status*** | | |
| Negative | 36,726 | 94.7 (94.5, 94.9) |
| Positive | 2,064 | 5.3 (5.1, 5.5) |
| **Number of ANC visits*** | | |
| ≥4 | 33,905 | 67.9 (67.5, 68.3) |
| <4 | 16,006 | 32.1 (31.7, 32.5) |
| **Premature rapture of the membranes (PROM)** | | |
| No | 49,770 | 97.9 (97.8, 98.0) |
| Yes | 1,077 | 2.1 (2.0, 2.2) |
| **Postpartum Hemorrhage (PPH)** | | |
| No | 50,572 | 99.5 (99.4, 99.5) |
| Yes | 275 | 0.5 (0.5, 0.6) |
| **Abruption placenta** | | |
| No | 50,676 | 99.7 (99.6, 99.7) |
| Yes | 171 | 0.3 (0.3, 0.4) |
| **Placenta previa** | | |
| No | 50,740 | 99.8 (99.7, 99.9) |
| Yes | 107 | 0.2 (0.2, 0.3) |
| **Gestational age (weeks) *** | | |
| Term birth (≥37) | 41,646 | 88.9 (88.6, 89.2) |
| Preterm birth (<37) | 5,184 | 11.1 (10.8, 11.4) |
| **Delivery mode*** | | |
| Vaginal | 33,526 | 66.1 (65.7, 66.5) |
| CS | 17,179 | 33.9 (33.5, 34.3) |
| **Referred for delivery*** | | |
| Yes | 11,488 | 23.5 (23.1, 23.8) |
| No | 37,479 | 76.5 (76.2, 76.9) |
| **Birth weight*** | | |
| NBW | 45,269 | 89.3 (89.0, 89.5) |
| LBW | 5,445 | 10.7 (10.5, 11.0) |
| **Sex of the baby*** | | |
| Male | 26,159 | 51.7 (51.2, 52.1) |
| Female | 24,461 | 48.3 (47.9, 48.8) |

*Frequencies do not tally to the total due to missing values in these variables

**Table 3. Distribution of missing values in the KCMC medical birth registry, 2000–2015 (N = 50,847).**

| Variables with missing data | Frequency | Percent Missing |
|---|---|---|
| Body Mass Index (BMI) | 15,911 | 31.3 |
| HIV status | 12,057 | 23.7 |
| Gestational age at birth categories | 4,017 | 7.9 |
| Referral status | 1,880 | 3.7 |
| Antenatal Care visits | 936 | 1.8 |
| Alcohol use during pregnancy | 636 | 1.3 |
| Occupation | 308 | 0.6 |
| Sex of the child | 227 | 0.5 |
| Marital Status | 186 | 0.4 |
| Mode of delivery | 142 | 0.3 |
| Birth weight of the child | 133 | 0.3 |
| Perinatal status (primary outcome) | 123 | 0.2 |
| Area of residence | 120 | 0.2 |
| Education level | 100 | 0.2 |
| Age categories | 85 | 0.2 |

(equivalent to a perinatal mortality rate (PMR) of 41.6 (95%CI 39.9, 43.3) deaths per 1,000 births). After the imputation of missing values, the proportion of perinatal death remained relatively the same. The proportion of perinatal death was significantly different (P<0.05) across maternal characteristics during pregnancy and delivery except for malaria, placenta previa, and sex of the child. Among deliveries from pre-eclamptic/eclamptic mothers, 13.6% ended up in a perinatal death. About 5% of perinatal deaths were from HIV positive mothers, 3.7% among those who drank alcohol during pregnancy, and 6.4% among those with <4 ANC visits. Deliveries from mothers who experienced postpartum hemorrhage (PPH) had high (22.3%) prevalence of perinatal death compared to those who were not (4.1%), p<0.001. Like-wise, the proportion was 58.8%, 15.6%, 19.1%, and 8.4% among deliveries from mothers who experienced abruption of placenta, preterm birth, low birth weight (LBW), and those referred for delivery, respectively (Table 4).

## Trends of perinatal death from 2000 to 2015 in northern Tanzania

Between 2000 and 2015, perinatal deaths have been declining slightly in this cohort (Fig 2). The proportion of perinatal death decreased significantly by 12.4% (95%CI 8–16.9%, p<0.001) for every one-year increase. The proportion of stillbirths is higher than that of early neonatal deaths over the years (Fig 3). However, between the years 2013–2015, the proportion of early neonatal deaths increased while that of stillbirth decreased, from 0.2% and 3.7% in 2013 to 1.8% and 2.2% in 2015, respectively. The proportion of stillbirth decreased significantly by 6.2% (95%CI 1.7–10.6%, p = 0.0.01) while that of early neonatal death decreased by 12.4% (95%CI 0.4–12.4%, p = 0.04) for every one-year increase.

## Predictors of perinatal death

Adjusted analysis for the predictors of perinatal death before and after imputation is in Table 5. Ignoring missing values in the analysis of pregnancy-related outcomes, especially in registry-based studies, produces biased parameter estimates. The coefficients resulting from complete case analysis are observed to be either higher or lower than should have been if there were no missing values. At the same time, standard errors are all relatively larger. For instance,

**Table 4. Perinatal status by maternal characteristics during pregnancy and delivery (N = 50,847).**

| Characteristics | Perinatal status | | p-value |
| --- | --- | --- | --- |
| | Alive | Died | |
| **Pre-eclampsia/eclampsia** | | | <0.001 |
| No | 46,836 (96.2) | 1,827 (3.8) | |
| Yes | 1,780 (86.4) | 281 (13.6) | |
| **Anaemia** | | | 0.001 |
| No | 47,875 (95.9) | 2,057 (4.1) | |
| Yes | 741 (93.6) | 51 (6.4) | |
| **Malaria** | | | 0.45 |
| No | 42,045 (95.9) | 1,811 (4.1) | |
| Yes | 6,571 (95.7) | 297 (4.3) | |
| **Infections** | | | 0.17 |
| No | 47,779 (95.8) | 2,080 (4.2) | |
| Yes | 837 (96.8) | 28 (3.2) | |
| **HIV Status** | | | 0.001 |
| Negative | 35,369 (96.5) | 1,296 (3.5) | |
| Positive | 1,959 (95.1) | 101 (4.9) | |
| **Drank alcohol during this pregnancy** | | | 0.01 |
| Yes | 13,593 (96.3) | 525 (3.7) | |
| No | 34,436 (95.7) | 1,538 (4.3) | |
| **Number of ANC visits** | | | <0.001 |
| ≥4 | 32,840 (97.1) | 976 (2.9) | |
| <4 | 14,947 (93.6) | 1,029 (6.4) | |
| **PROM** | | | 0.004 |
| No | 47,569 (95.8) | 2,082 (4.2) | |
| Yes | 1,047 (97.6) | 26 (2.4) | |
| **PPH** | | | <0.001 |
| No | 48,403 (95.9) | 2,047 (4.1) | |
| Yes | 213 (77.7) | 61 (22.3) | |
| **Abruption placenta** | | | <0.001 |
| No | 48,546 (96.0) | 2,008 (4.0) | |
| Yes | 70 (41.2) | 100 (58.8) | |
| **Placenta previa** | | | 0.21 |
| No | 48,517 (95.8) | 2,101 (4.2) | |
| Yes | 99 (93.4) | 7 (6.6) | |
| **Gestational age (weeks)** | | | <0.001 |
| Term birth (≥37) | 40,483 (97.4) | 1,064 (2.6) | |
| Preterm birth (<37) | 4,362 (84.4) | 805 (15.6) | |
| **Delivery mode** | | | 0.32 |
| Vaginal | 32,078 (95.8) | 1,396 (4.2) | |
| CS | 16,435 (96.0) | 682 (4.0) | |
| **Referred for delivery** | | | <0.001 |
| Yes | 10,487 (91.6) | 967 (8.4) | |
| No | 36,365 (97.2) | 1,033 (2.8) | |
| **Birth weight** | | | <0.001 |
| NBW | 44,132 (97.7) | 1,037 (2.3) | |
| LBW | 43,92 (80.9) | 1,035 (19.1) | |
| **Sex of the baby** | | | 0.88 |

(*Continued*)

**Table 4.** (Continued)

| Characteristics | Perinatal status | | p-value |
|---|---|---|---|
| | Alive | Died | |
| Male | 25,018 (95.9) | 1,082 (4.1) | |
| Female | 23,408 (95.9) | 1,006 (4.1) | |
| **Total** | **48,616 (95.8)** | **2108 (4.2)** | |

considering Model II, there was a reduced risk of perinatal death among deliveries of adolescent mothers from (RR = 0.582, 95%CI 0.478, 0.708) to (RR = 0.674, 95%CI 0.575, 0.789) compared to those aged 20–34 years before and after imputation, respectively. Also, the risk of perinatal death was observed to have reduced from (RR = 1.432 1.243, 1.648) to (RR = 1.423, 1.263, 1.603) among pre-eclamptic/eclamptic mothers and increased from (RR = 2.072, 95%CI 1.847, 2.324) to (RR = 2.111, 1.906, 2.338) among preterm deliveries, before and after imputation, respectively. Consumption of alcohol during pregnancy lowered the risk of perinatal death except in Model II after imputation. While the direction of the association for both Models I and II is almost the same; results from Model II provide more precise estimates of the predictors

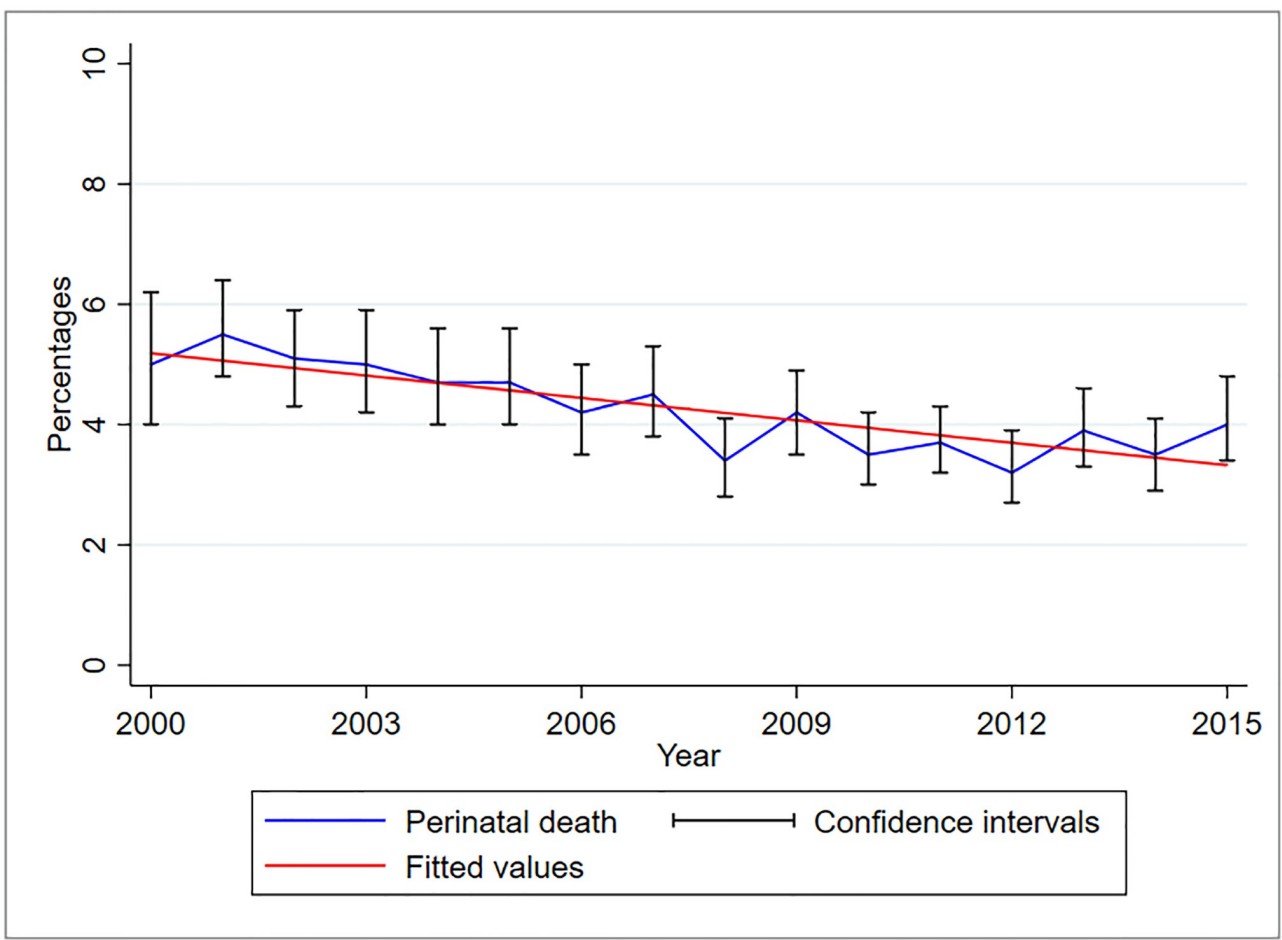

**Fig 2. Trends of perinatal deaths, KCMC medical birth registry data, 2000–2015 (N = 50,724).**

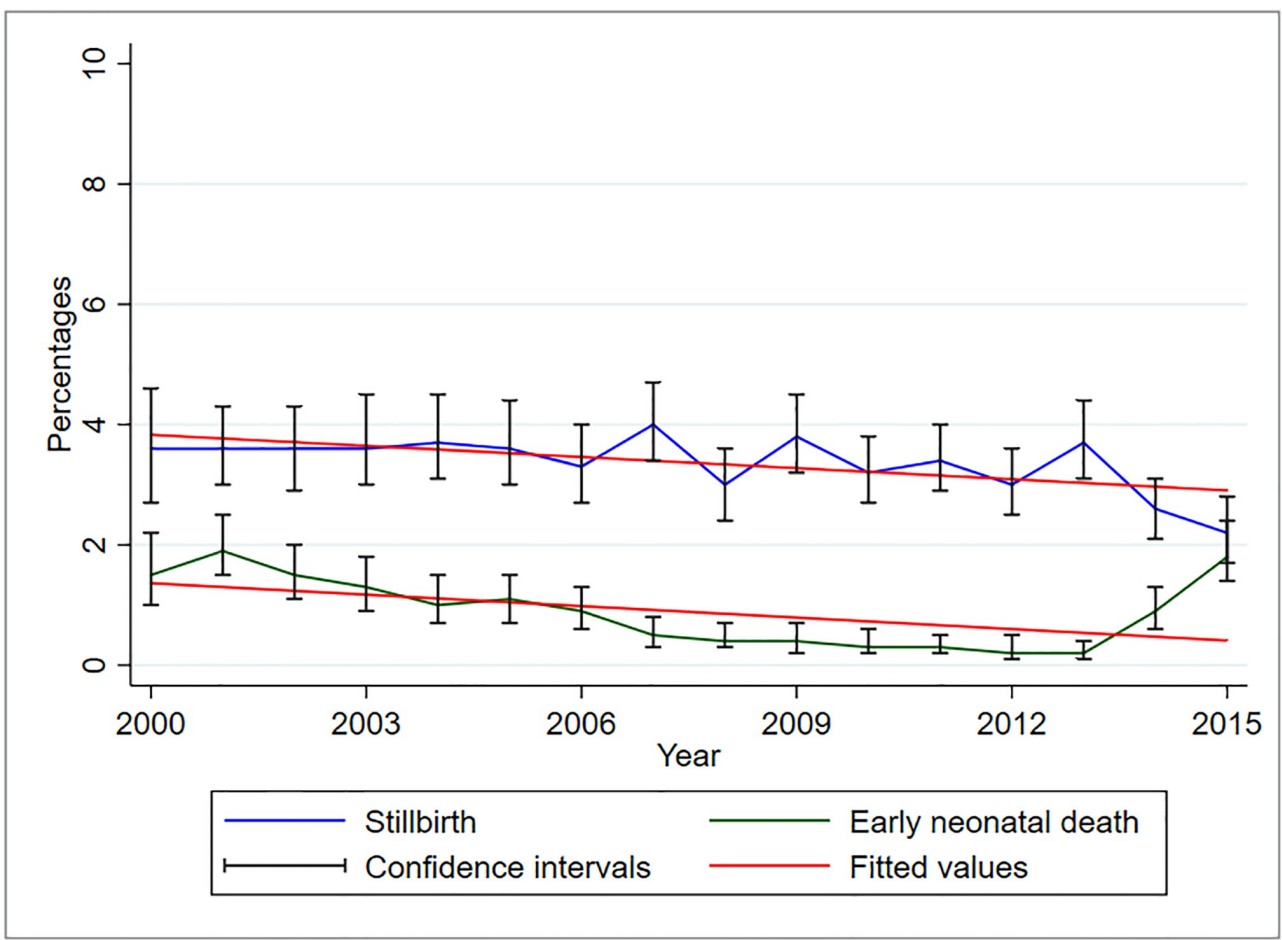

**Fig 3. Trends of stillbirths and early neonatal deaths, KCMC medical birth registry data, 2000–2015 (N = 50,724).**

of perinatal death as it has both considered missing values and accounted for overdispersion of the data. The precision of estimates is not only affected by missing data but also the type of regression model used for analysis. For instance, results for the association between abruption placenta and risk of perinatal death in Model I before and after imputation of missing values shows a nearly 20 times higher risk of perinatal death and too wide confidence intervals. Model II, however, provide more precise estimates both before and after imputation (Table 5). We will, therefore, focus on results for Model II for interpretation of our findings.

Based on Model II, the risk of perinatal death was higher among deliveries from mothers who resided in rural compared to urban areas (RR = 1.241, 95%CI 1.137, 1.355), with primary (RR = 1.201, 95%CI 1.083, 1.332) compared to higher education levels, with <4 ANC visits (RR = 1.250, 95%CI 1.146, 1.365) compared to ≥4 visits, with PPH (RR = 2.638, 95%CI 1.997, 3.486), abruption placenta (RR = 4.218, 95%CI 3.438, 5.175), LBW (RR = 4.210, 95%CI 3.788, 4.679), male child (RR = 1.090, 95%CI 1.007, 1.181), and were referred for delivery (RR = 2.108, 95%CI 1.919, 2.317). On the other hand, a lower risk of perinatal death was observed among deliveries from mothers who experienced PROM (RR = 0.411, 95%CI 0.283, 0.598) and delivered through CS (RR = 0.662, 95%CI 0.604, 0.724).

We further examined the effect of eight or more ANC visits on the risk of perinatal death as recommended by WHO [17]. We would like to indicate it here that data analyzed in this study

**Table 5. Adjusted analysis showing predictors of perinatal death.**

| Characteristics | BEFORE IMPUTATION (N = 43,198) | | | | AFTER IMPUTATION (N = 50,847) | | | |
| --- | --- | --- | --- | --- | --- | --- | --- | --- |
| | Model I[a] | | Model II[b] | | Model I[a] | | Model II[b] | |
| | AOR[c] (SE) | 95%CI | ARR[d] (SE) | 95%CI | AOR[c] (SE) | 95%CI | ARR[d] (SE) | 95%CI |
| **Age groups (years)** | | | | | | | | |
| 15–19 | 0.524 (0.059) | 0.419, 0.654*** | 0.582 (0.058) | 0.478, 0.708*** | 0.614 (0.058) | 0.510, 0.738*** | 0.674 (0.054) | 0.575, 0.789*** |
| 20–34 | 1.000 | | 1.000 | | 1.000 | | 1.000 | |
| 35–39 | 1.277 (0.102) | 1.092, 1.493** | 1.196 (0.081) | 1.048, 1.365** | 1.281 (0.090) | 1.116, 1.471*** | 1.201 (0.070) | 1.072, 1.347** |
| 40+ | 1.375 (0.197) | 1.039, 1.821* | 1.320 (0.150) | 1.056, 1.650* | 1.439 (0.178) | 1.129, 1.834** | 1.348 (0.131) | 1.114, 1.630** |
| **Area of residence (Rural)** | 1.221 (0.072) | 1.087, 1.371*** | 1.202 (0.061) | 1.088, 1.328*** | 1.274 (0.067) | 1.150, 1.412*** | 1.241 (0.056) | 1.137, 1.355*** |
| **Highest education level** | | | | | | | | |
| None | 1.291 (0.254) | 0.878, 1.898 | 1.182 (0.212) | 0.832, 1.679 | 1.548 (0.223) | 1.167, 2.054** | 1.401 (0.171) | 1.103, 1.780** |
| Primary | 1.224 (0.082) | 1.073, 1.397** | 1.211 (0.071) | 1.080, 1.358** | 1.216 (0.075) | 1.078, 1.372** | 1.201 (0.063) | 1.083, 1.332*** |
| Secondary | 1.013 (0.109) | 0.821, 1.251 | 1.024 (0.094) | 0.855, 1.226 | 0.956 (0.095) | 0.787, 1.161 | 0.975 (0.082) | 0.827, 1.150 |
| Higher | 1.000 | | 1.000 | | 1.000 | | 1.000 | |
| **Pre-eclampsia/eclampsia (Yes)** | 1.638 (0.156) | 1.359, 1.975*** | 1.432 (0.103) | 1.243, 1.648*** | 1.642 (0.136) | 1.396, 1.932*** | 1.423 (0.087) | 1.263, 1.603*** |
| **Drank alcohol during this pregnancy (Yes)** | 0.860 (0.058) | 0.755, 0.981* | 0.886 (0.051) | 0.791, 0.991* | 0.889 (0.053) | 0.791, 0.998* | 0.910 (0.046) | 0.825, 1.004 |
| **Number of ANC visits (<4)** | 1.313 (0.076) | 1.172, 1.470*** | 1.267 (0.064) | 1.148, 1.399*** | 1.294 (0.067) | 1.169, 1.433*** | 1.250 (0.056) | 1.146, 1.365*** |
| **PROM (Yes)** | 0.322 (0.082) | 0.195, 0.530*** | 0.377 (0.088) | 0.238, 0.597*** | 0.346 (0.074) | 0.228, 0.525*** | 0.411 (0.079) | 0.283, 0.598*** |
| **PPH (Yes)** | 4.299 (1.014) | 2.708, 6.825*** | 2.648 (0.435) | 1.918, 3.654*** | 4.781 (0.993) | 3.183, 7.182*** | 2.638 (0.375) | 1.997, 3.486*** |
| **Abruption placenta (Yes)** | 19.469 (4.515) | 12.357, 30.673*** | 4.472 (0.525) | 3.552, 5.629*** | 18.349 (3.821) | 12.200, 27.598*** | 4.218 (0.440) | 3.438, 5.175*** |
| **Gestational age (<37 weeks)** | 2.416 (0.161) | 2.121, 2.752*** | 2.072 (0.122) | 1.847, 2.324*** | 2.495 (0.148) | 2.222, 2.802*** | 2.111 (0.110) | 1.906, 2.338*** |
| **Delivery mode (CS)** | 0.569 (0.036) | 0.502, 0.645*** | 0.634 (0.034) | 0.571, 0.704*** | 0.594 (0.034) | 0.531, 0.664*** | 0.662 (0.031) | 0.604, 0.724*** |
| **Birth weight (LBW)** | 5.064 (0.336) | 4.446, 5.768*** | 4.243 (0.258) | 3.765, 4.781*** | 5.118 (0.302) | 4.560, 5.745*** | 4.210 (0.227) | 3.788, 4.679*** |
| **Sex of the baby (Male)** | 1.119 (0.062) | 1.005, 1.247* | 1.115 (0.052) | 1.017, 1.222* | 1.098 (0.054) | 0.998, 1.208 | 1.090 (0.045) | 1.007, 1.181* |
| **Referred for delivery (Yes)** | 2.577 (0.161) | 2.280, 2.912*** | 2.218 (0.120) | 1.994, 2.466*** | 2.465 (0.139) | 2.207, 2.754*** | 2.108 (0.101) | 1.919, 2.317*** |
| **Year** | 0.954 (0.007) | 0.941, 0.967*** | 0.962 (0.006) | 0.951, 0.973*** | 0.950 (0.006) | 0.939, 0.961*** | 0.960 (0.005) | 0.951, 0.969*** |

[a]GEE model with binomial family, logit link, exchangeable correlation and robust variance estimator;

[b] GEE log-linear regression model, i.e., Poisson family, log link, an exchangeable correlation, and robust variance estimator;

[c] Adjusted Odds Ratio;

[d] Adjusted Risk Ratio;

*p<0.05,

**p<0.01,

***p<0.001

are between the years 2000 and 2015, during which the recommendation was at least four ANC visits, but WHO issued the new guidelines in 2016. About 94% of 49,911 recorded deliveries had <8 ANC visits. Results from the complete case analysis of the GEE log-linear model (Poisson family, log link function, exchangeable correlation and robust variance estimator) almost agreed with those presented in Table 5 for the four or more ANC visits. The risk of perinatal death among deliveries from women with <8 ANC visits in this model was (RR = 1.372, 95%CI 1.031, 1.825) compared to ≥8 ANC visits. The relative risk of perinatal death in the same model (Model II in Table 5, in the complete case analysis) for <4 visits is 1.267 (95%CI 1.148, 1.399), though slightly lower.

## Stratified analysis by referral status

Deliveries from women referred for delivery in this cohort had a higher risk of experiencing perinatal death (RR = 2.577, 95%CI 2.280, 2.912) compared to those who were not (Table 5). These women are at risk of having serious pregnancy complications [3] that are likely to increase the risk of perinatal death. We, therefore, performed stratified analysis (of the imputed data) by referral status to better understand the risk factors of perinatal death (Table 6). Findings of the GEE log-linear regression model (i.e., Poisson family, log link,

**Table 6. Adjusted analysis showing predictors of perinatal death stratified by referral status.**

| Characteristics | Referred for delivery (N = 11,488) | | Not referred for delivery (N = 37,479) | |
|---|---|---|---|---|
| | ARR [a] (SE) | 95%CI | ARR [a] (SE) | 95%CI |
| **Age groups (years)** | | | | |
| 15–19 | 0.619 (0.065) | 0.503, 0.761*** | 0.667 (0.093) | 0.508, 0.875** |
| 20–34 | 1.000 | | 1.000 | |
| 35–39 | 1.250 (0.104) | 1.062, 1.471** | 1.233 (0.100) | 1.051, 1.445* |
| 40+ | 1.273 (0.172) | 0.977, 1.660 | 1.407 (0.211) | 1.049, 1.886* |
| **Area of residence (Rural)** | 1.131 (0.075) | 0.993, 1.288 | 1.257 (0.076) | 1.117, 1.414*** |
| **Highest education level** | | | | |
| None | 1.249 (0.194) | 0.921, 1.694 | 1.511 (0.297) | 1.028, 2.220* |
| Primary | 1.115 (0.967) | 0.941, 1.321 | 1.229 (0.085) | 1.072, 1.408** |
| Secondary | 0.999 (0.128) | 0.777, 1.285 | 0.923 (0.110) | 0.731, 1.165 |
| Higher | 1.000 | | 1.000 | |
| **Pre-eclampsia/eclampsia (Yes)** | 1.339 (0.113) | 1.135, 1.580** | 1.588 (0.141) | 1.335, 1.890*** |
| **Drank alcohol during this pregnancy (Yes)** | 1.152 (0.084) | 0.998, 1.329 | 0.757 (0.053) | 0.659, 0.869*** |
| **Number of ANC visits (<4)** | 1.357 (0.088) | 1.194, 1.541*** | 1.175 (0.074) | 1.039, 1.330* |
| **PROM (Yes)** | 0.426 (0.107) | 0.261, 0.697** | 0.243 (0.089) | 0.118, 0.500*** |
| **PPH (Yes)** | 3.182 (0.521) | 2.308, 4.387*** | 1.816 (0.470) | 1.094, 3.015* |
| **Abruption placenta (Yes)** | 4.024 (0.521) | 3.121, 5.186*** | 5.023 (0.734) | 3.772, 6.689*** |
| **Gestational age (<37 weeks)** | 1.687 (0.129) | 1.451, 1.960*** | 2.523 (0.201) | 2.158, 2.949*** |
| **Delivery mode (CS)** | 0.628 (0.039) | 0.556, 0.708*** | 0.651 (0.048) | 0.563, 0.752*** |
| **Birth weight (LBW)** | 2.722 (0.203) | 2.351, 3.151*** | 6.220 (0.492) | 5.328, 7.263*** |
| **Sex of the baby (Male)** | 0.983 (0.059) | 0.875, 1.105 | 1.156 (0.068) | 1.030, 1.297* |
| **Year** | 0.979 (0.007) | 0.965, 0.994** | 0.946 (0.007) | 0.932, 0.959*** |

Stratified analysis was performed after imputing the missing data (conditional on each referral status) using the GEE log-linear regression model, i.e., Poisson family, log link, exchangeable correlation, and robust variance estimator.

[a] Adjusted Risk Ratio;

*p<0.05,

**p<0.01,

***p<0.001

exchangeable correlation and robust variance estimator) indicated that, the risk of perinatal death among those referred for delivery was significantly higher among deliveries from women aged 35–39 years (RR = 1.250, 95%CI 1.062, 1.471) compared to those aged 15–19 years, with preeclampsia/eclampsia (RR = 1.339, 95%CI 1.135, 1.580), <4 ANC visits (RR = 1.357, 95%CI 1.194, 1.541), with PPH (RR = 3.182, 95%CI 2.308, 4.387), abruption placenta (RR = 4.024, 95% CI 3.121, 5.186), preterm birth (RR = 1.687, 95%CI 1.451, 1.960) and delivered a LBW baby (RR = 2.722, 95%CI 2.351, 3.151) compared to those who did not. PROM and CS delivery were still protective against perinatal death, (RR = 0.426, 95%CI 0.261, 0.697) and (RR = 0.628, 95% CI 0.556, 0.708), respectively. Maternal area of residence, education level, alcohol use during pregnancy and sex of the child were not associated with the risk of perinatal death in this analysis. Although not statistically significant, deliveries from women who consumed alcohol during pregnancy had 1.152 (95%CI 0.998, 1.329) times the risk of experiencing a perinatal death.

It is worth noting here that, compared to women referred for delivery, there is a stronger association between most of the covariates and the risk of perinatal death in the group of women not referred for delivery (Table 6). Higher risk of perinatal death is still on deliveries from women aged 35–39 years, and 40+ years compared to 20–34 years, rural residents, those with low education level, with pre-eclampsia/ eclampsia, <4 ANC visits, PPH, Abruption play-center, preterm birth, LBW, and male child. Deliveries from women aged 15–19 years, with PROM and delivered through CS had lower risk of experiencing a perinatal death (Table 6). It was noted that under Model II after imputation of missing values, consumption of alcohol during pregnancy among women not referred for delivery had a strong protective effect on the risk of perinatal death (RR = 0.757, 95%CI 0.659, 0.869, p<0.001).

## Predictors of stillbirth

The proportion of stillbirths in this study is higher than that of early neonatal deaths as shown in Fig 3. To better understand the risk factors for death among neonates in this cohort, we excluded 415 (0.8%) early neonatal deaths among 50,847 total deliveries. We then reanalyzed the data after performing multiple imputation of missing values. It can be observed (in Table 7) that, results from this analysis (the GEE log-linear model–Poisson family, log link, exchangeable correlation, and robust variance estimator) agrees with those for the predictors of perinatal deaths in Table 5 except for the sex of the child and consumption of alcohol during pregnancy. The risk of perinatal death is significantly higher among deliveries from women aged 35–39 (RR = 1.252, 95%CI 1.103, 1.421) and 40+ (RR = 1.366, 95%CI 1.096, 1.701) compared to 20–34 years, rural residents (RR = 1.273, 95%CI 1.153, 1.405), with no education (RR = 1.428, 95%CI 1.089, 1.873) and primary education level (RR = 1.224, 95%CI 1.090, 1.374) compared to higher education. The risk is also high among deliveries from women with pre-eclampsia/ eclampsia (RR = 1.497, 95%CI 1.311, 1.710), <4ANC visits (RR = 1.286, 95%CI 1.163, 1.423), PPH (RR = 3.100, 95%CI 2.282, 4.210), abruption placenta (RR = 4.683, 95%CI 3.722, 5.893), preterm birth (RR = 2.210, 95%CI 1.961, 2.492), delivered a LBW baby (RR = 4.619, 95%CI 4.091, 5.214) and were referred for delivery (RR = 2.080, 95%CI 1.870, 2.313). The risk of stillbirth decreased significantly with every one-year increase in time (RR = 0.970, 95%CI 0.960, 0.980). Furthermore, the risk of perinatal death was low among deliveries from women aged 15–19 years (RR = 0.574, 95%CI 0.475, 0.695), who experienced PROM (RR = 0.302, 95%CI 0.184, 0.497), and delivered through CS (RR = 0.555, 95%CI 0.449, 0.617).

## Discussion

The proportion of perinatal death in this study was 4.2%, an estimated PMR of 41.6 per 1000 births. This study found that, under MAR assumption, ignoring missing values leads to biased

**Table 7. Adjusted analysis showing predictors of stillbirth (N = 50,432).**

| Characteristics | ARR[a] (SE) | 95%CI |
|---|---|---|
| **Age groups (years)** | | |
| 15–19 | 0.574 (0.056) | 0.475, 0.695*** |
| 20–34 | 1.000 | |
| 35–39 | 1.252 (0.081) | 1.103, 1.421*** |
| 40+ | 1.366 (0.153) | 1.096, 1.701** |
| **Area of residence (Rural)** | 1.273 (0.064) | 1.153, 1.405*** |
| **Highest education level** | | |
| None | 1.428 (0.198) | 1.089, 1.873* |
| Primary | 1.224 (0.072) | 1.090, 1.374** |
| Secondary | 0.957 (0.091) | 0.795, 1.152 |
| Higher | 1.000 | |
| **Pre-eclampsia/eclampsia (Yes)** | 1.497 (0.101) | 1.311, 1.710*** |
| **Drank alcohol during this pregnancy (Yes)** | 0.909 (0.051) | 0.815, 1.015 |
| **Number of ANC visits (<4)** | 1.286 (0.066) | 1.163, 1.423*** |
| **PROM (Yes)** | 0.302 (0.077) | 0.184, 0.497*** |
| **PPH (Yes)** | 3.100 (0.484) | 2.282, 4.210*** |
| **Abruption placenta (Yes)** | 4.683 (0.549) | 3.722, 5.893*** |
| **Gestational age (<37 weeks)** | 2.210 (0.135) | 1.961, 2.492*** |
| **Delivery mode (CS)** | 0.555 (0.030) | 0.499, 0.617*** |
| **Birth weight (LBW)** | 4.619 (0.286) | 4.091, 5.214*** |
| **Sex of the baby (Male)** | 1.081 (0.050) | 0.988, 1.183 |
| **Referred for delivery (Yes)** | 2.080 (0.113) | 1.870, 2.313*** |
| **Year** | 0.970 (0.005) | 0.960, 0.980*** |

Analysis performed using the GEE log-linear regression model, i.e., Poisson family, log link, exchangeable correlation, and robust variance estimator.

[a] Adjusted Risk Ratio;

*p<0.05,

**p<0.01,

***p<0.001

parameter estimates. Higher risk of perinatal death was associated with maternal demographic characteristics (i.e., age, area of residence and education level), pregnancy and delivery-related characteristics (i.e., pre-eclampsia/eclampsia, number of ANC visits, PPH, abruption placenta, preterm birth, low birth weight, sex of the child and referral status). PROM and CS delivery were associated with a lower risk of perinatal death in this population. There were no differences in the predictors of perinatal death and stillbirth in this study.

The PMR in this study 41.6 per 1000 births is higher than 27.0 per 1,000 births reported in Manyara region, northern Tanzania[40], and slightly higher than the national estimate, 39 per 1000 births [2] but significantly higher than those reported from high-income countries such as Ireland (5.4/1,000 births)[11] and USA (6.0/1,000 births)[41]. Higher PMR than the national level could be because this study was conducted at the consultant referral hospital and therefore attending women with most complications or higher risk pregnancy compared to estimates at lower-level facilities and population surveys. PMR in this study is lower than those reported in India (49.4/1,000 births), Nigeria (49.9/1,000 births)[42, 43], and in Eastern Sudan (75.3 per 1,000 births)[44]. Higher rate in Sudan was linked to the higher proportion of home deliveries, low ANC coverage, and maternal infections such as malaria and anemia[44]. These

inequalities reflect differences in the availability and quality of obstetric and newborn care services within and between countries. The proportion of stillbirth in this study was consistently larger than the early neonatal death, possibly because women and their newborns were discharged soon after birth; hence deaths occurring at home could not be recorded[45]. However, between the years 2013–2015, the proportion of early neonatal deaths increased while that of stillbirth decreased. A previous study on the recording of maternal deaths reported a considerable under-reporting of these deaths in the medical birth registry [46], which might also be the case in the reporting of stillbirth and early neonatal deaths in this study. Concerted efforts are needed to articulate and address the health system and provided related challenges in the delivery of quality reproductive, maternal, and newborn care services to avert the rising trends of PMR in Tanzania.

Ignoring missing values in the analysis of clinical events such as adverse pregnancy outcomes like perinatal death produces biased parameter estimates [19–24], as also shown in this study. After the imputation of missing values, the effect of covariates on perinatal death appeared to be stronger than before imputation. This emphasizes the need for considering missing values in modeling adverse pregnancy outcomes, particularly in longitudinal studies where such issues are common[22, 24, 35]. We further observed that, the GEE log-binomial regression model slightly over-estimated parameter estimates compared to the log-linear model. For more precise parameter estimates, data analysts should carefully consider the choice of the regression models that better fit their data.

Unexpectedly, alcohol use during pregnancy was observed to reduce the risk of perinatal death before imputation of missing values and after imputation using the GEE binomial regression model. We did not find a significant association in the GEE with a log-linear regression model, which agrees with other studies[47]. However, maternal alcohol consumption during pregnancy has been associated with a decreased risk of being small for gestational age (below 10$^{th}$ percentile related to gestational age and sex) and that of preterm birth[48]. Although the stratified analysis in our study agrees with these findings only among women not referred for delivery, this finding should be interpreted with caution due to the low proportion (3.5%) of women who drank alcohol during pregnancy in this category. Moreover, the study did not collect data on the dose-effect relationship for the association between alcohol consumption and perinatal death. More should be done to investigate the effect of alcohol use during pregnancy on the adverse pregnancy outcomes because it is unlikely that alcohol consumption in itself can explain these findings [48]. Furthermore, Isaksen et. al., [48] recommended further insights on diet (and nutrition), lifestyle factors, and maternal health that could explain this this paradoxical association. Despite the observed association, the WHO recommends screening and counseling of pregnant women on the harmful effects of substance use during pregnancy (including alcohol) at every ANC visit[17]. We found the risk of perinatal death to be high among deliveries from mothers in higher age categories (35–39 and 40+ years) compared to those aged 20–34 years. Advanced maternal age (>35 years) has been linked with a higher risk of stillbirth[49], while in Uganda, there was no significant age effect [10]. However, the risk of perinatal death was high among pregnant women aged 30+ years[10], which can be explained by small number of perinatal deaths in the 30+ age category. Rural residents had a higher risk of experiencing perinatal death in this study, contrary to findings from Uganda, who found the risk to be high in urban areas[10]. Women from urban areas in Uganda were living in slums (characterized by low socio-economic status (SES) and poor access to care), which could explain the observed discrepancies. Furthermore, compared to our findings, there was no significant association between perinatal death and maternal education level in this study[10]. It is worth noting here that, having no education was not associated with the risk of perinatal death before but not after imputation of missing values. Pregnant women residing in rural areas and with low education

levels could be in a disadvantaged position in accessing quality health care during pregnancy and childbirth. Low socioeconomic status has also been linked to adverse pregnancy outcomes such as perinatal death[42, 49, 50], but this could not be measured in our study.

Area of residence and education level were not significant predictors of perinatal death among those referred for delivery probably because most of those referred were coming from rural areas of residence, had low SES and low education. Deliveries from women who had <4 ANC visits had a higher risk of perinatal death. Antenatal care provides a critical opportunity for women and babies to benefit from good quality maternal care[16, 17]. Furthermore, the risk of perinatal death was higher among males compared to female children in this study, which also agrees with findings from Brazil[51]. This may be linked to early pulmonary maturation among females that lowers the risk of respiratory complications[51], one of the leading causes of under-five deaths in low-income countries[13]. However, sex of the child was not associated with the risk of perinatal death and stillbirth among women referred for delivery. Interventions to increase coverage and uptake of recommended routine ANC services for pregnant women are crucial for early identification and management of pregnancy-related complications.

The risk of perinatal death was higher among deliveries from women with hypertensive disorders of pregnancy particularly pre-eclampsia/eclampsia, those with postpartum hemorrhage, abruption placenta, delivered preterm and low birth weight baby, which is similar to other studies[4, 12, 42, 49]. We did not find a significant association between maternal anemia, malaria infection, positive HIV status, systemic infections/sepsis, and placenta previa with an increased risk of perinatal death. PROM and delivery by CS reduced the risk of perinatal death in this cohort. The latter is known to reduce the risk of obstetric complications when medically indicated[2, 12]. The protective effect of PROM could reflect timely management of such pregnancies considering that these deliveries were attended at a tertiary care facility where comprehensive emergency obstetric and newborn care (CEmONC) services are available. Despite that, a study in Uganda has shown that CS delivery, particularly among women with PROM, increases the risk of perinatal mortality among other adverse pregnancy outcomes [52].

Women referred for delivery are potentially at risk of experiencing adverse maternal and perinatal outcomes [3, 12, 14]. In this study, we included referral cases in our analyses and later performed a stratified analysis. As one would expect, the risk of perinatal death was almost twice higher among deliveries from women referred for delivery compared to those who were not. This is because these women are likely to have experienced pregnancy-related complications that required specialized care[3, 12]. Findings from the stratified analysis by referral status were almost comparable with non-referral results except for area of residence, education level, alcohol use during pregnancy, and sex of the child, which were not statistically significant, in those referred. Yet, we cannot ignore the fact that women referred for delivery are most at risk of experiencing adverse pregnancy outcomes. It is essential to strengthen the referral system to ensure timely and proper referral mechanisms and to promote appropriate health care seeking behavior to reduce the risk of perinatal deaths[3]. At the same time, we observed a stronger association between covariates and the risk of perinatal death in the group of women not referred for delivery compared to those referred. This group of women should also be given due attention in order to prevent the rise in perinatal death cases.

Being a registry-based study from a zonal referral hospital, findings (on predictors of perinatal death) may not be generalized to a larger population. However, population-based estimates agree with these results[10, 40, 44]. Furthermore, the fact that this birth registry only captured deaths occurring in the health facility lowers the number of recorded neonatal deaths. It could also underestimate the proportion and rates of perinatal death. Hence, stillbirth contributed to a higher percentage of the perinatal death numbers. This could explain the

unobserved differences between predictors of perinatal death and stillbirth in this study. To our knowledge, no study has assessed the effect of ignoring missing values on determining predictors of adverse pregnancy outcomes, particularly perinatal death. This study has provided evidence for the need to consider missing values in the analysis of pregnancy outcomes. Our findings also emphasize the necessity of proper choice of statistical models for more precise parameter estimates. Increased surveillance and management of different maternal risk factors for different perinatal outcomes should be at the heart of improving child survival.

## Acknowledgments

We would like to acknowledge the midwives who participated in data collection and all women and children whose information enabled the availability of data used in this study. The authors also thank the staff at the Birth Registry for capturing these data in the electronic system. We also appreciate the Centre for International Health at the University of Bergen in Norway and the Department of Obstetrics and Gynecology of the KCMC hospital in Tanzania for establishing the KCMC medical birth registry, which facilitated the availability of data to conduct this study.

## Author Contributions

**Conceptualization:** Innocent B. Mboya, Michael J. Mahande, Joseph Obure, Henry G. Mwambi.

**Data curation:** Innocent B. Mboya, Michael J. Mahande.

**Formal analysis:** Innocent B. Mboya, Henry G. Mwambi.

**Methodology:** Innocent B. Mboya, Joseph Obure, Henry G. Mwambi.

**Project administration:** Michael J. Mahande, Joseph Obure, Henry G. Mwambi.

**Supervision:** Michael J. Mahande, Joseph Obure, Henry G. Mwambi.

**Validation:** Innocent B. Mboya, Michael J. Mahande, Joseph Obure, Henry G. Mwambi.

**Visualization:** Innocent B. Mboya, Henry G. Mwambi.

**Writing – original draft:** Innocent B. Mboya.

**Writing – review & editing:** Innocent B. Mboya, Michael J. Mahande, Joseph Obure, Henry G. Mwambi.

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
