## [Decision Letter · Decision Letter 0]

2 Mar 2020

PONE-D-19-33574

Predictors of perinatal death in the presence of missing data: A registry-based study in northern Tanzania

PLOS ONE

Dear Mr. Mboya,

Thank you for submitting your manuscript to PLOS ONE. After careful consideration, we feel that it has merit but does not fully meet PLOS ONE’s publication criteria as it currently stands. Therefore, we invite you to submit a revised version of the manuscript that addresses the points raised during the review process.

We would appreciate receiving your revised manuscript by Apr 16 2020 11:59PM. To enhance the reproducibility of your results, we recommend that if applicable you deposit your laboratory protocols in protocols.io, where a protocol can be assigned its own identifier (DOI) such that it can be cited independently in the future. For instructions see: http://journals.plos.org/plosone/s/submission-guidelines#loc-laboratory-protocols

We look forward to receiving your revised manuscript.

Kind regards,

Angela Lupattelli, PhD

Academic Editor

PLOS ONE

2. Please amend your current ethics statement to address the following concerns: Please explain why was written consent was not obtained, how you recorded/documented participant consent, and if the ethics committees/IRBs approved this consent procedure.

"We would like to acknowledge the midwives who participated in data collection as well as all

women whose information enabled availability of data used in this study. We also acknowledge

the Norwegian Council for Higher education program and Development Research (NUFU) for

funding the establishment of the KCMC Medical birth registry. This funding was not related to

this manuscript.".

"This work was supported through the DELTAS Africa Initiative Grant No. 107754/Z/15/Z-DELTAS Africa SSACAB. The DELTAS Africa Initiative is an independent funding scheme of the African Academy of Sciences (AAS)’s Alliance for Accelerating Excellence in Science in Africa (AESA) and supported by the New Partnership for Africa’s Development Planning and Coordinating Agency (NEPAD Agency) with funding from the Wellcome Trust (Grant No. 107754/Z/15/Z) and the UK government. The views expressed in this publication are those of the author(s) and not necessarily those of AAS, NEPAD Agency, Wellcome Trust or the UK government.

The funders had no role in study design, data collection and analysis, decision to publish, or preparation of the manuscript.".

Reviewers' comments:

Reviewer's Responses to Questions

**Comments to the Author**

1. Is the manuscript technically sound, and do the data support the conclusions?

Reviewer #1: Partly

Reviewer #2: Partly

2. Has the statistical analysis been performed appropriately and rigorously? 

Reviewer #1: No

Reviewer #2: No

3. Have the authors made all data underlying the findings in their manuscript fully available?

Reviewer #1: No

Reviewer #2: No

4. Is the manuscript presented in an intelligible fashion and written in standard English?

Reviewer #1: Yes

Reviewer #2: Yes

5. Review Comments to the Author

Reviewer #1: Thank you for the invitation to review this manuscript, which I read with great interest. The authors used data from KCMC medical birth register, which collected information from mothers about their health during pregnancy, delivery details and characteristics of the child. The authors used these data to explore risk factors for perinatal deaths (stillbirths and early neonatal deaths), applying multiple imputation to account for missing data. The paper looks at an interesting subject, but some key information about missing data is not reported (especially given it’s the key focus of the paper), and I have some concerns about selection of variables in the model. Major revisions are required prior to publication.

Major comments:

Given that evaluation of the impact of missing data is one of the key aims of the paper, there is insufficient information about missing data, patterns and imputation. The authors should consider adding information on:

1. the patterns of missing data: how much data was missing and for which variables? the authors state in tables 1-2 that counts don’t add up to the total, but the number of pregnancies with missing data should be added. It is not clear how many children were included in the cohort overall and how many in the complete case cohort for analyses?

2. what was the impact of missing data on the perinatal mortality? It is not clear what was the perinatal mortality rate in the full cohort (including all pregnancies with missing data) compared to complete case cohort. It is unclear why the mortality rate went up after imputation (from 41.6/1000 births to 53/1000 births)

3. What was the imputation model? Which variables needed to be imputed and which were used to predict missing data?

4. Authors state in multiple parts of the manuscript that complete case analyses were biased and led to incorrect conclusions. However, the results seem to be largely consistent between complete case and imputed data, with slightly different effect sizes, but the same conclusions would be reached based on complete case data. I think such strong statements should be avoided

The authors might find these suggested reporting guidelines for papers using multiple imputation useful: Sterne JA, White IR, Carlin JB, et al. Multiple imputation for missing data in epidemiological and clinical research: potential and pitfalls. BMJ. 2009;338:b2393. Published 2009 Jun 29. doi:10.1136/bmj.b2393

My second major comment related to the modelling strategy. What was variable selection strategy? It looks that all risk factors available were included in the model, but they are likely to be collinear, eg having various pregnancy complications is probably closely correlated with referral for delivery. Having multicollinearity in the model could introduce bias to the analysis, e.g. the odd result for alcohol consumption during pregnancy (showing protective effect) could be a result of that over adjustment in presence of other unmeasured confounding. The authors should have a look at literature on birthweight paradox.

Other comments:

1. Perinatal mortality should be defined as stillbirths and early neonatal deaths per 1000 total births, including live and stillbirths (see WHO ICD10 instruction manual https://www.who.int/classifications/icd/ICD10Volume2_en_2010.pdf?ua=1), but the authors report perinatal mortality per 1000 live births

2. Abstract: explain all abbreviations as it is not obvious what they are

3. Methods: Why did the authors run models to estimate both odds ratios and risk ratios, rather than just risk ratios?

4. Methods: The authors note that mothers could chose not to participate - what proportion of mothers opted out and hence is not covered?

5. Results: are figures 1 and 2 showing percentage out of all births? Since the authors have fitted a trend, by how much did perinatal mortality decline? The results splitting perinatal mortality into stillbirths and early neonatal deaths could be discussed more – the declines were driven by early perinatal mortality as stillbirth rate remained constant, but in most recent years something has changed and early neonatal deaths went up while stillbirth went down. Why is that? Is it due to registration practices? It would be worth discussing.

Reviewer #2: Reviewed comments from Unnati Saha

The manuscript “Predictors of perinatal death in the presence of missing data: A registry-based study in northern Tanzania”

The paper focused on an important topic to address on the determinants of perinatal mortality and importance of missing observations and their attribution in the analysis. My comments are given below.

-Overall the paper is written well, but additional revision by an expert English editor is necessary for the improvement of the manuscript for publication in Plos One.

Specific comments are given as below.

Abstract:

Backgorund

-perhaps it would be complete if author(s) replace “zonal”with the complete name of the hospital e.g., Kilimanjaro Christian Medical centre (KCMC).

Methods

-In this section Author(s) probably can mention KCMC instead writing the full name of the hospital

Conclusion

- “Appropriate choice of the analysis model should be considered in similar studies”. Not clear about this sentence.

- Which one is benchmark model? Please refer the benchmark model based on results

- Also, better if author(s) revise the last two sentences under the conclusion section.

Main Body text

Overall comments

- Manuscript does not include page numbers and line numbers, which hinders to reporting reviewed comments orderly

Introduction:

- paragraph third, last sentence, I understand that the author(s) performed analysis with and without missing values imputation and compared the results for statistical significance. So, perhaps it would be better if they explain the aim of the study completely.

Methods:

Study design and participants

-second paragraph, Author(s) did not make it explicit why they excluded multiple births from the analysis

-the explanation of the method how missing values were imputed are missing, also better give a theoretical explanation of the use of two methods how one model better perform over other model.

variables

- author(s) included so many variables in the analysis, so I am wondering whether it hampers the effects of some variables on estimated perinatal mortality due to co-linearity. For example, low birthweight, gestational week, preterm birth, and other medical indicators. So, author(s) need to be cautious about entering variables in the model.

-also, as WHO recommended for 8 ANC visits (I read in the INTRODUCTION section), author(s) perhaps could try to investigate the effects of 8 visits as well.

- I miss the definition of the perinatal death (as dependent variable) whether dummy or continuous, although next section statistical analysis it gives a hint.

Statistical analysis

- first paragraph, the explanation of the method how missing values were imputed are missing.

- Model I (‘binomial family’ means outcome variable is coded as dummy), Model II (log-linear regression with passion family) means outcome variable is count variable. How they handle the 0 counts (no perinatal deaths) in the analysis.

- Author(s) performed two models using two types of outcomes. Author(s) could play with variables in different models of both outcome variables “perinatal mortality”, e.g., child, mother, family, clinical etc. different levels of variables. The results could provide better understanding of the effects on the model outcome.

- author(s) applied GEE method, but as author(s) expect multiple deaths born to a mother, so how the analysis handled the correlated deaths in one family (mother)?

- if deaths are correlated and not accounted for this, it may spurious the conclusion on determinants of perinatal deaths. Author, may give explanation on this.

- again, perhaps author(s) can explain imputation technique in better way for general readers.

Results:

Maternal background and characteristics at first birth

- Why first birth? I understand that author(s) analyzed perinatal mortality that occurred at any birth born to a mother in the hospital and also in rural area referred to this hospital

- Few cases (<=.5%) are recorded against variables PPH, Abruption placenta, placenta previa, so I am wondering whether entering those variables in the model will be appropriate. It may miss-specify the model. Also, better if there is an explanation how those variables can effect pregnancy and mortality e.g., PROM.

- Also check all variables distribution including alcohol drinking for the referral cases as analysis was stratified by referral vs not referral cases.

- Table 3, should it be “maternal characteristics”? revise the title

Predictors of perinatal death

-Table 4, n should be different before and after imputation, so better to mention them

- do the author(s) have any explanation how alcohol consumption may reduce perinatal deaths (see Table 4)? Also, explain the results of alcohol consumption in model 5 for referral cases vs. not referral cases.

-Table 5, which model employed for this analysis? Seeing the sample size it seems model with imputation data, but not clear which model (from the title)? Model I or Model II?

- Seeing Table 5 and coefficients of alcohol use it seems to me that better exclude this variable from the analysis. The effects are different for referral and not referral, and that is the reason that in Model II in Table 4, after imputation the effect was lower and insignificant.

- Table 6, again correct the number of cases

- Women had complication and that is why probably they had CS delivery. It’s effect may affect the effects of referral including other complication variables.

- Better to add a section explaining model and better fitted model

- Discussion section requires careful revision with proper explanation of the effects with references and conclusion drawn from this study.

I would suggest to accept the paper after incorporating the suggested comments indicated as above of this note.

6. PLOS authors have the option to publish the peer review history of their article (what does this mean?). If published, this will include your full peer review and any attached files.

Reviewer #1: No

Reviewer #2: No

---

## [Author Response · Author response to Decision Letter 0]

26 Mar 2020

All responses to the editorial and reviewer comments are contained in the cover letter titled "Response to Reviewers" as part of the attachments.

We have also provided the email for the executive director of the KCMC referral hospital, to which data request can be submitted. The email is also provided here; drgmasenga@yahoo.com.

---

## [Editor Report · Decision Letter 1]

30 Mar 2020

Predictors of perinatal death in the presence of missing data: A registry-based study in northern Tanzania

PONE-D-19-33574R1

Dear Dr. Mboya,

We are pleased to inform you that your manuscript has been judged scientifically suitable for publication and will be formally accepted for publication once it complies with all outstanding technical requirements.

With kind regards,

Angela Lupattelli, PhD

Academic Editor

PLOS ONE
---

## [Editor Report · Acceptance letter]

2 Apr 2020

PONE-D-19-33574R1 

Predictors of perinatal death in the presence of missing data: A registry-based study in northern Tanzania 

Dear Dr. Mboya:

I am pleased to inform you that your manuscript has been deemed suitable for publication in PLOS ONE. Congratulations! Your manuscript is now with our production department. 

With kind regards,

on behalf of

Dr. Angela Lupattelli 

Academic Editor

PLOS ONE